# Synthesis of Prussian Blue Nanoparticles and Their Antibacterial, Antiinflammation and Antitumor Applications

**DOI:** 10.3390/ph15070769

**Published:** 2022-06-21

**Authors:** Danyang Li, Meng Liu, Wenyao Li, Qiang Fu, Liyang Wang, Enping Lai, Weixin Zhao, Kaile Zhang

**Affiliations:** 1The Department of Urology, Affiliated Sixth People’s Hospital, Shanghai Jiao Tong University, Shanghai 200233, China; lidanyang19940@163.com (D.L.); memgqinlawliet@126.com (M.L.); jamesqfu@126.com (Q.F.); 2School of Materials Engineering, Shanghai University of Engineering Science, Shanghai 201620, China; m050119110@sues.edu.cn; 3Guangxi Key Laboratory of Green Processing of Sugar Resources, College of Biological and Chemical Engineering, Guangxi University of Science and Technology, Liuzhou 545026, China; nemodhu@163.com; 4Wake Forest Institute of Regenerative Medicine, Winston Salem, NC 27101, USA; wl2xzhao@gmail.com

**Keywords:** Prussian blue nanoparticles, tissue repair, anti-inflammatory effects, nano-enzyme, drug carrier

## Abstract

In recent years, Prussian blue nanoparticles (PBNPs), also named Prussian blue nano-enzymes, have been shown to demonstrate excellent multi-enzyme simulation activity and anti-inflammatory properties, and can be used as reactive oxygen scavengers. Their good biocompatibility and biodegradability mean that they are ideal candidates for in vivo use. PBNPs are highly efficient electron transporters with oxidation and reduction activities. PBNPs also show considerable promise as nano-drug carriers and biological detection sensors owing to their huge specific surface area, good chemical characteristics, and changeable qualities, which might considerably increase the therapeutic impact. More crucially, PBNPs, as therapeutic and diagnostic agents, have made significant advances in biological nanomedicine. This review begins with a brief description of the synthesis methods of PBNPs, then focuses on the applications of PBNPs in tissue regeneration and inflammation according to the different properties of PBNPs. This article will provide a timely reference for further study of PBNPs as therapeutic agents.

## 1. Introduction

Prussian blue (PB) was discovered in the early 18th century as a coordination polymer [1]. In 1936, powder diffraction started to be used to investigate the crystal structure of PB, and then, in 1977, single crystal diffraction was used [2,3]. PB is also known as iron ferrocyanide for its excellent gas storage, metal ion fixation, proton conduction, irritation-affected magnetism, ease of synthesis and considerable electrochemical properties [4,5,6]. The co-existence of Fe^3+^ and Fe^2+^ ions gives PB excellent magnetic, electrical and optical properties, which have been extensively studied in the fields of batteries, sensors and catalysis [7,8,9,10]. In the field of electrochemistry, PB is an excellent electron transfer medium. Fe ions with different valence states within the molecule can transfer electrons internally through the cyanide group, and the PB’s absorption spectrum reaches the maximum at 680 nm [1]. A simple synthesis process may be used to make colloid PB; the electron transfer from the Fe^2+^ core to the Fe^3+^ center leads to the blue color [11].

With their excellent characteristics such as multienzyme-like activities [12], biocompatibility, biodegradability, controllable shape and size, low production cost, and easy synthesis, Prussian blue nano-particles (PBNPs), also known as Prussian blue nano-enzymes (PBzyme), have been widely used in biomedical fields, such as targeted drug delivery [13]. It becomes an ideal molecular carrier for diagnosis and therapy to improve the efficiency of drug delivery for targeted therapy by adjusting its morphology [14].Moreover, the excellent biosafety and biocompatibility of PBNPs provide a strong basis for its application as a nano drug carrier and imaging agent [15]. In 2003, the US Food and Drug Administration approved PBNPs as a treatment for acute and chronic Thallium poisoning [16], which will have good prospects in the development of nanodrugs for disease diagnosis and treatment.

The development of nanotechnology has had an impact on the field of biomedical diagnosis and treatment, thanks to the development of new biomedical functions of traditional materials and the nanocrystalization of traditional drugs [17]. In order to prepare nano drugs using a simple process at a low cost, with a stable chemical structure, good dispersibility, strong durability and mass production, the development of PB in biomedical applications will be beneficial to improve the safety of existing medical preparation biological technology, which is currently not guaranteed. So, it has great value in research application.

In recent years, the discovery of nano-enzyme activity has further promoted the application of PBNPs in antibacterial, anti-inflammatory, tissue repair and regeneration, antitumor and ultrasound imaging. In this review, we first introduce the crystal structure of PBNPs and then describe the various classical synthetic strategies of PBNPs including different morphologies and doping. We highlight the antibacterial, anti-inflammatory and antitumor applications of PBNPs, especially the properties of PBNPs as a multienzyme activity of PBzyme to prevent vascular restenosis and to treat Parkinson’s disease. We also focus on the application of PBNPs as a carrier for antitumor drugs and tumor and cell detection. Some technical challenges faced by PBzyme in clinical application are also analyzed, and the possible research directions to overcome these challenges are proposed. The objective is to achieve clinical treatment by local injection of PBzyme to accomplish disease treatment by the minimally invasive or even non-invasive combination of PBzyme, and to provide a basis for promoting further research and development of PBNPs.

## 2. Crystal Structure, Preparation and Characterization of PBNPS

### 2.1. Crystal Structure of PBNPs

PB and its analogues (PBA) are a variety of microporous inorganic solids, and the unit lattice structure of PBA is shown in (Figure 1A) [18]. The structure is formed by the coordination of Fe^2+^ and Fe^3+^ ions with a cyanide group (−CN^−^) [7]. PB containing Fe^3+^ and Fe^3+^[Fe^3+^(CN)_6_] can be oxidized to Prussian yellow or be reduced to Prussian white to become Fe^2+^ and Fe^2+^[Fe^2+^(CN)_6_]^2−^ due to the presence of the mixed-valence Fe in PB. PB has two crystalline forms, insoluble and soluble (Figure 1B) [14].

PBNPs have a crystalline framework structure mainly composed of Fe^3+^, Fe^2+^ and bridging cyano groups. To build a three-dimensional coordination network, Fe^3+^ and Fe^2+^ ions are alternatively organized with bridging cyanides. Among them, the Fe^2+^ cations are coordinated by the carbon atom of the cyanide (CN^−^) ligand, which acts as a bridge with Fe^3+^ cations which are octahedrally coordinated by 6 nitrogen atoms. In the unit cell, the dimensions are 10.2 Å, and the average bond lengths of Fe(II)-C = 1.92 Å, C-N = 1.13 Å, and Fe(III)-N = 2.03 Å, respectively [3]. PB is insoluble due to its three-dimensional coordination network structure, which is also why PB exists as nanoparticles. Soluble and insoluble PB are so-called because soluble PB produces lower dimensioned crystals that reach the size typical of the mesophase, i.e., nanoparticles, forming the blue colloidal solutions. In contrast, insoluble PB tends to aggregate and form precipitates that can form larger crystals [19].

### 2.2. Synthesis Method of PBNPs

With the development in synthesis technology, the preparation of PB with a specific composition, different morphology and different particle size becomes more mature. At present, the preparation methods of PBNPs include the single and double precursor methods [20], reversed phase microemulsion method [21], hydrothermal method [22], and so on. Different synthesis methods have their own advantages and disadvantages; now, most of them are able to prepare the uniform and stable single crystal PBNPs.

Yang et al. developed the single precursor synthesis method first in 1998 [20]. The ferricyanide complex of K_3_[Fe(CN)_6_] or K_4_[Fe(CN)_6_] ions is usually used as the precursor, and Fe^3+^or Fe^2+^ ions can be slowly released in acidic solution, either reduced or oxidized to Fe^2+^/Fe^3+^. Compared with the two-precursor synthesis, the reaction time of the single precursor is longer, but the control is stronger, and the sample is more uniform, but it will produce trace toxicity and cannot be synthesized on a large scale. In the double precursor synthesis process, the equal molar amount of Fe^3+^/Fe^2+^ and [Fe(CN)_6_]^4−^/[Fe(CN)_6_]^3−^ is directly mixed to form the PBNPs. The advantages of this synthesis method are short reaction time and no requirement of reducing agent, but its dispersibility, stability and morphology control need to be further improved [23]. In 2000, Vaucher et al. synthesized (Cu_2_Fe(CN)_6_) in microemulsion medium [21]. They limited the synthesis to nanoscale water droplets formed in inverted microemulsion and prepared hydrophobic PBNPs of the same shape and size. The advantage of this method is that the morphological regulation is stronger, but it does not explain the exact cause of restrictions on nanoparticles. At the same time, some researchers compared the single precursor method with the reverse phase microemulsion method [24]. It was found that the single precursor method was simpler and convenient, but the dispersion stability in the physiological environment was still poor. To solve this problem, researchers have adopted different methods to prepare PBNPs. For example, the hydrothermal method has the advantages of being a simple process, low cost, and easy for large preparation. It can introduce in situ modification based on PVP and K_3_[Fe(CN)_6_] to prepare PBNPs with at least 90 days of physiological stability [22]. For example, PBNPs were synthesized by reacting FeCl_2_ and FeCl_3_ with K_3_Fe(CN)_6_, respectively [25]. The particle size distribution obtained by Fe^2+^ was narrower (1.7 nm), but the stability was poor (Figure 2A). The particle size obtained by Fe^3+^ was relatively stable but decreased exponentially (18 nm). The one-step hydrothermal method for producing monodisperse PVP functionalized PBzyme [26] has a diameter of about 67 nm and is stable for at least 90 days (Figure 2B). OH-scavenging capability was substantial at 50 ppm in system studies, and the absorbance of the peroxidase substrate rose at 650 nm. In general, the catalase (CAT) and superoxide dismutase (SOD) activities of PBzyme were often extremely high. These methods provide important prerequisites for its further clinical application in the future. So, to better apply PBNPs to the biomedical field, researchers [27] obtained a hollow PB(HPB) nano-enzyme (HPBzyme), with size and particle size controllable by hydrothermal template-free vacuum freeze-drying (Figure 2C). The mesoporous structure of PBzymes has a high specific surface area and pore volume, which can be used as a drug carrier. It can be applied in conjunction with other therapies and play a larger role in treatment.

Through the summary and comparison of different synthetic methods, it was found that the reaction time of the nanoparticles obtained is shorter, the morphological regulation is stronger, and the scattered stability is better, and the final structure and particle size can be controlled. Additionally, the application is more practical and diversified.

### 2.3. Synthesis of PBNPs with Different Morphologies

PB is a kind of classical Fe^2+^ and Fe^3+^ mixed valence hexyanoferrite. The preparation method is simple, and the particle size is easy to control with surface modification. Since the single crystal PBNPs prepared have relatively simple properties, researchers prepared mesoporous PBNPs through surface–specific modification and morphology regulation to obtain a drug delivery system with a high drug load and doped other ions to make it more abundant to practical development prospects. Owing to continuous development, PBNPs with different morphologies can be synthesized by different methods under different experimental conditions.

Lee et al. etched PB with different concentrations of nitric acid. This method revealed that the internal plane of PB crystal has different stability, so various cubes can be prepared [28]. Wang et al. synthesized cubic, spherical, different sizes (~30–200 nm) and positively charged PBNPs by changing the rate of iron ions and adjusting the reaction time (Figure 3A) [15]. Lu et al. prepared HPB nanospheres, which solved the problem that the shell thickness and size of PBNPs could not be controlled; they also studied in vitro and in vivo biological effects in detail [29]. Zhang et al. synthesized HPBZs with internal cavities through a simple, Bi^3+^-assisted, template-free synthesis strategy (Figure 3B) [30]. This method has the advantages of being a simple process with high physiological stability and no need for any post-treatment. The natural enzyme has high catalytic efficiency, high sensitivity and continuous detection, but the cost is expensive and unstable with the appearance of artificial nano-enzyme. Wang et al. synthesized Molybdenum-polysulfide-deposited nickel-iron bimetal PB-analog-based hollow Nanocages, which showed significantly higher activity than the original PB analogue, and stronger stability under extreme conditions [31]. These results indicate that PBNPs can be designed to biomedicine according to demand.

A number of studies have shown that the morphology, size, absorption cross-sectional area, sharpness and density of nanoparticles all have great influence on their photothermal properties under certain external experimental conditions. The morphology of PBNPs is closely related to the photothermal properties, and the photothermal properties are different from the particle morphology, leading to the occurrence of different biological effects. PBNPs have multi-enzyme imitate activity and are also called PBzyme. PBzyme is a kind of nanomaterial with activity similar to the natural enzyme [32,33]. As a consequence of their excellent catalytic efficiency, stability, low cost, and ease of modification, these emerging artificial enzymes have been attracting the attention of biomedical researchers [34].

Due to the open lattice structure of PBNPs crystals, they can absorb and capture other molecules or ions in the lattice holes, thus introducing other molecules or metal ions. Moreover, in practical studies such as promoting tissue regeneration, eliminating inflammation and preventing vascular restenosis, many PBNPs have been doped to enhance their biological and chemical properties. The use of PBNPs alone has limitations in enhancing or inhibiting biological activity by controlling doping ratios and acting on different signaling pathways. At present, partial doping of PBNPs mainly includes Mn^2+^ [35], Zn^2+^ [36], Ag [37], Gd^3+^ [38], FA [39], Pt^2+^ [40], PVP [41], VANNPs [42], whose functions will be described in the following paragraphs.

## 3. Application of Enzyme Activity of PBNPs

### 3.1. Anti-Inflammatory Effects

Inflammation is a natural immune response to injury or infection. Reactive oxygen species (ROS) are derived from metabolic processes in organisms and are an important intermediate in cell signal transduction and intracellular functions. The presence of a number of ROS is the key to inflammation, which may provide a new method for the treatment of inflammation [41]. PBNPs have REDOX and enzyme activities and are a nano-enzyme with various catalytic properties, which can effectively scavenge ROS [43]. In addition, a complete substitution (PBA) and partial substitution (doping) allow the introduction of new substances and improve the performance of PBNPs. In treating diseases caused by ROS, it is effective to activate the intrinsic biological activity of a single component nano-enzyme and modify the microenvironment. Therefore, many researchers have used this method to further study the inhibition of PBNPs on various inflammation in humans. The role of PBNPs in various inflammatory processes is reviewed (Figure 4), as shown in the followings.

#### 3.1.1. Role in the Prevention of Vascular Restenosis

The destruction of the vascular endothelial cell barrier system leads to vascular restenosis [44,45], which is also a difficult problem for intravascular interventional therapy and seriously affects the long-term and prognostic quality of patients [45,46].

A new discovery suggests that PBzyme can effectively scavenge ROS and accumulate significantly in damaged vascular intima and macrophages [26]. Vascular balloon injury (VBI)-induced endothelial cell exfoliation can enhance the passive infiltration of nano-enzymes to improve vascular restenosis. A diagram of nano-enzymes preventing vascular restenosis is shown in (Figure 5B). M1 macrophages exacerbate the inflammation and vascular restenosis by interfering with the recovery of endothelial cells. In contrast, M2 macrophage polarization prevents vascular restenosis by reducing smooth muscle cell migration and proliferation, reducing secretion of proinflammatory cytokines and promoting endothelial cell growth.

In addition, Feng et al. reported a mesoporous PBzyme coated with neutrophil-like cell membrane (NCM) (MPBzyme @NCM) that can actively target MPBzyme @NCM delivery to the injured brain by binding to the inflamed brain microvascular endothelial cells. MPBzyme @NCM accumulated in the injured brain is engulfed by microglia and has a long-term effect on ischemic stroke (Figure 5A) [47]. Detailed mechanisms include microglia polarization to M2, reduced neutrophil recruitment, neuronal apoptosis, proliferation of neural stem cells, neuronal precursors, and neuronal cells. This research method will provide a new application prospect for the treatment of various brain diseases with nano-enzymes. Ischemic stroke is a devastating disease. After the ischemic injury, excessive production of reactive oxygen and nitrogen species (RONS) leads to an aggravation of brain injury. Zhang et al. constructed HPBzyme through Bi^3+^-assisted, template-free synthesis strategies that drive neuroprotection against ischemic stroke by scavenging RONS (Figure 5C) [30]. In addition to alleviating oxidative stress, HPBzyme can also inhibit apoptosis and inflammatory response in vivo and in vitro, thus improving the tolerance of cerebral ischemia injury with minimal side effects. This study provides evidence for novel neuroprotective nanoagents that may benefit the treatment of ischemic stroke and other diseases associated with RONS. In order to facilitate the clinical transformation of HPBzyme, further research is needed, including the exploration of the blood–brain barrier (BBB) penetrable modification, the long-term therapeutic effects of injecting HPBzyme in posterior brain mode, and the long-term toxicity.

#### 3.1.2. Treatment of Parkinson’s Disease

Current pharmacological interventions for Parkinson’s disease (PD) remain unsatisfactory in clinical settings. Inflammasome-mediated pyroptosis agonists or antagonists are expected to have positive effects on neurodegenerative diseases. PBzyme was discovered to be a pyroptosis inhibitor, alleviating neurodegeneration in animal and cell models of PD and protecting microglia and neurons against 1-methyl4-phenyl-1,2,3,6-tetrahydropyridine (MPTP) [48]. The experimental findings demonstrate that PBzyme might suppress microglia pyroptosis by downregulating gasdermin D (GSDMD) cracking and inflammatory factor generation. Scavenging ROS reduce the activation of microglial nucleotide-binding domain and leucine-rich repeat family pyrin domain containing 3 (NLRP3) inflammasomes and caspase-1. Although this discovery reveals a significant strategy for the use of PBNPs in the therapy of neurodegenerative disorders, the pharmacokinetics and long-term toxicity of PBzyme in vivo are currently unclear.

PD is mainly caused by mitochondrial dysfunction, which produces ROS and abnormal energy metabolism. Liu et al. synthesized a zeolite imidazole framework 8 coated with PB nanocomposite (ZIF-8@PB), which was coated with the antioxidant quercetin (QCT) [49]. In MPTP-induced mice models, ZIF-8@PB-QCT exhibited outstanding near-infrared radiation (NIR) response by activating the PI3K/Akt signaling pathway, penetrating the BBB directly to the site of injury under the guidance of the photothermal effect, and realizing intracranial drug delivery for the treatment of neurodegenerative diseases. Due to its good biocompatibility, this non-invasive strategy is promising as an intracerebral delivery route for treating neurodegenerative diseases.

#### 3.1.3. Treatment of Inflammatory Bowel Disease

Inflammatory bowel disease (IBD) is an inflammatory illness in which the immune system destroys digestive system cells, and the overproduction of ROS is critical to the course of IBD and may be a potential therapeutic target. Zhao et al. created poly (vinylpyrrolidone)-modified PBNPs (PPBs) with high physiological stability and biosafety using a one-pot technique to overcome PB insolubility [41]. The produced PPBs have the capacity to remove ROS and suppress pro-inflammatory cytokines and may considerably decrease colitis in mice following intravenous delivery with no noticeable adverse effects. Prior to this 2018 study, nanomedicine based on PBNPs had not been investigated or used in the treatment of IBD. In 2019, Zhao et al., for the first time, synthesized Manganese Prussian blue nano-enzymes (MPBZs) by mixing manganese source solution with PVP and [Fe(CN)_6_]^4−^ source solution with PVP through a simple hydrothermal method. The mice colitis model induced by sodium dextran sulfate (DSS) was established [50]. In mice with IBD, MPBZs with multi-enzyme activity accumulated preferentially on the mucosal surface of colitis inflammation in positively charged mice through a size/charge mediated inflammation targeting strategy. MPBZs can operate as a new generation of inflammation-targeting ROS due to their low redox potential and abundance of varying valence states of Mn^2+^ and Fe^2+^. Since elevated levels of ROS affect cell proliferation and lead to aging and apoptosis, excessive ROS accumulation at the wound site can lead to inflammation, necrosis and fibrosis scarring of skin cells, and ultimately delay the tissue repair process [12,51,52,53]. In addition, MPBZs reduce the incidence of colitis in mice by acting on the TLR signaling pathway and have fewer side effects.

#### 3.1.4. Treatment of Osteoarthritis

Osteoarthritis (OA) microenvironment disrupts ROS balance, extracellular matrix synthesis, and chondrocyte degradation. The intervention of biological and mechanical variables in the middle and early stages of osteoarthritis can successfully prevent disease progression and avoid joint replacement. Hou et al. designed HPBzyme by the hydrothermal method [27]. In a rat model, HPBzyme was found to greatly protect chondrocytes and delay the onset of traumatic OA by rebuilding the microenvironment by reducing ROS and Rac1/nuclear factor Kappa-B (NF-B) signaling. HPBzyme decreased IL-1-stimulated inflammation, extracellular matrix breakdown, and human chondrocyte apoptosis substantially. This study indicates that the exploitation of biological activity of nanomedicine is worthy of attention in the diagnosis and treatment of major diseases.

ROS-induced chondrocyte apoptosis and extracellular matrix (ECM) destruction are critical in the pathophysiology of osteoarthritis. PBNPs can scavenge ROS in cells. Zuo et al. initially applied PBNPs combined with Low-Intensity Ultrasound Alleviates (LIPUS) to activate the PI3K/Akt/mTOR pathway to reduce chondrocyte ROS and apoptosis in the treatment of knee osteoarthritis (KOA) [54]. PBNPs/LIPUS combination therapy reduced ROS, apoptosis, matrix metalloproteinases (MMPs) and expression of inflammatory cytokines, and inhibited ECM degradation. It can be used as a non-invasive treatment for OA rehabilitation, but the clinical efficacy of PBNPs/LIPUS needs to be further studied.

#### 3.1.5. Treatment of Acute Pancreatitis and Peritonitis

Acute pancreatitis (AP) has a high morbidity and mortality due to the lack of specific clinical drugs and surgical treatment. It is known that the pancreatic enzyme activity is activated and the NF-κB signaling pathway is activated [55], leading to acute pancreatitis, e.g., pancreatic tissue digestion, edema, bleeding, and necrosis [56,57,58,59]. Xie et al. prepared PBzyme by polyvinylpyrrolidone modification and verified the anti-inflammatory and scavenging effects of PBzyme at the cellular level [60]. PBzyme delays AP by inhibiting the TLRs/NF-κB signaling pathway associated with inflammation and oxidative stress, scavenging reactive oxygen species, and stimulating the antioxidant and anti-inflammatory biological activities of AP. This is compatible with the reported pharmacological mechanism for the treatment of AP, but PBzyme offers benefits such as ease of synthesis, convenient and steady internal biological action, biosafety, etc. Further research is required in the preventative effectiveness, long-term biosafety, and mechanism of PBzyme in AP non-human primate models. The findings of this study lay the groundwork for therapeutic use of PBzyme in AP and other ROS/inflammatory illnesses. Activation of hydrogen sulfide is an important cause of AP-related lung injury, with high mortality in cases. Liu et al. developed a novel pure-inorganic up-conversion nanoprobe using PB as the H_2_S responsive acceptor, which could be used for AP diagnosis and as a therapeutic agent for co-alleviating lung injury [61]. The 5 nm PB up-conversion nanoprobe (UC-PB_5_) improves the detection limit and range through optimization, and is used to detect and eliminate H_2_S, effectively inhibiting H_2_S-related MPO activation and further reducing oxidative stress in the lung and AP-related lung injury. This suggests a new model for the clinical diagnosis of AP, which would reduce mortality from lung injury and improve survival in AP cases.

Ansuja et al. developed hyaluronic acid-stabilized PBNPs (HAPB), enhancing the stability of nanoparticles and the uptake of inflammatory macrophages [62]. In mice models of peritonitis, the nanoparticles demonstrated strong peroxide-scavenging efficiency and decreased infiltration of inflammatory immune cells. HAPB nanoparticles decreased the number of M1 inflammatory macrophages but increased the amount of M2 anti-inflammatory markers. As a result, more research is required in the role of ROS in producing M1 macrophage markers and other variables that restrict the development of M2 macrophage markers. In conclusion, HAPB nanoparticles have high potential for therapeutic use in the treatment of inflammatory disorders caused by hydrogen peroxide.

Published data show that there are many prospects of PB for the treatment of inflammation. However, nano-enzyme drug research is not mature, now rarely used in clinical trials. In order to overcome this defect, eventually it asks for medical researchers to further accumulate data with the goal of applying PBzyme in clinical settings.

### 3.2. Anti-bacteria and Promote Wound Healing

Bacterial infections have long been a vexing medical problem and a huge economic burden. It is of great significance to construct a multi-functional bacterial care point detection (POCT) and elimination platform efficiently. Typically, hydrogels or polymer-based materials are used to deliver drugs or various growth factors to promote wound healing and skin tissue regeneration [63]. In recent years, a variety of inorganic nanomaterials and nanotechnology-driven treatment methods have attracted wide attention in skin wound healing, and the repair effect of PBzyme on skin tissue has gradually entered the field of vision of researchers.

#### 3.2.1. Photothermal and Photodynamic Therapy

Traditional antibiotics do not target bacteria and cause general harm to human tissues [64,65]. Phototherapy is a potential treatment since it is non-invasive, does not cause medication resistance, and has low toxicity. In the presence of medicinal drugs, adequate light sources, and oxygen, it can cause bacterial harm by producing heat or reactive oxygen species [66,67,68]. Photodynamic therapy (PDT) and photothermal therapy (PTT) are classified according to different principles of laser therapy. Because of their depth of penetration, low toxicity, and excellent spatial/temporal resolution, near-infrared (NIR) absorbent photothermal agents have the potential to transform existing antibacterial techniques [69,70]. Most of the focus is to maximize the therapeutic effect and minimize the side effects under the coordination of PDT/PTT. Currently, treating bacterial infections only using photothermal treatment is difficult because the high temperatures necessary for bacterial eradication, although causing minimal harm to healthy tissue, can inevitably cause damage [71]. In 2019, Li et al. developed zinc-doped Prussian blue (ZnPB) as an exogenous antibacterial agent. ZnPB nanocubes with different doping levels were prepared by freeze drying [72]. It illustrates the mechanism of ZnPB-3 broad-spectrum bactericidal action (Figure 6A). Accelerating ion release and infiltration into bacteria under local thermal energy induced by photothermal effects, ZnPB was reported to de-activate the methicillin-resistant *S. aureus* (MRSA) in animal wound models, thereby altering cellular metabolic pathways and killing bacteria without systemic toxicity. In addition, the photothermal characteristics of ZnPB with varying doping levels were improved by controllable design by maximizing the antibacterial synergy of heat release and ion release. It was found that the synergy of local PTT and ion release in ZnPB-3 allows for the rapid removal of MRSA from the wound surface (Figure 6B). They investigated the appropriate temperature range for treating PTT with ZnPB-3 for 15 min and assessed the corresponding antibacterial efficiency and toxicity to healthy tissues (Figure 6C) [72]. The results show that when the maximum temperature of PTT was above 55 °C, the bacteriostasis rate reached more than 90% after 15 min. However, inflammatory responses were observed on neutrophils when PTT exceeded 60 °C after 15 min. This provides an effective reference range for future PTT evaluation. The principle of ZnPB treatment for bacterial wound infection is to promote collagen deposition and wound repair through the upregulation of tissue remodeling genes, high photothermal conversion efficiency and low laser flux. The combination of local light and heat, as well as the release of ions, provides a broad spectrum of antibacterial techniques, which enlightens the antibacterial application.

Most detection is based on surface-enhanced Raman scattering (SERS), which only focuses on bacteria detection and lacks bactericidal function. Gao et al. designed an interference-free SERS platform of sandwich structure with high sensitivity and accuracy, which could simultaneously and reliably detect bacterial colonies in whole blood samples and perform in situ photothermal elimination [73]. The SERS substrate was a plasmonic gold film (pAu) functionalized with bacteria-capturing units of 4-mercaptophenylboronic acid and an internal reference of 4-mercaptobenzonitrile. The vancomycin-modified core (gold)−shell PBNPs (Au@PB@Van NPs) was a SERS tag of platform (Figure 6D). Furthermore, the antibacterial efficacy is extremely excellent due to the photothermal synergy of pAu and Au@PB@Van NPs. It provides a new method of in situ sterilization and a new way for early diagnosis of serious diseases caused by bacterial infection.

To achieve rapid treatment of bacterial infection wounds, Luo et al. successfully prepared a Core-shell Dual Metal-Organic Framework (MOF) heterointerface, PB@ MOF, which is the core of growing porphyrin-doped MOF [74]. Although PB@MOF has both photothermal and photodynamic properties, its antibacterial effect is poor under single illumination (808 nm near infrared or 660 nm red light) for 10 min, and its antibacterial effect is greatly improved under dual illumination, indicating a good antibacterial effect from the synergy of photothermal and photodynamic effects. Sharma et al. synthesized a one-of-a-kind silver-doped PB nanoscale coordination polymer (SPB NCPs) in an aqueous solution of AgNO_3_ and potassium ferrocyanide (K_4_[Fe^II^(CN)_6_]) in the presence of citric acid (CA) for dual-mode photothermal ablations and bacterial cell oxidative toxicity [37]. SPB NCPs with Fe-CN-Ag bond was compared with PB. It was found that SPB NCP and PB had similar physical size, crystal phase and optical properties. However, only SPB NCPs showed oxidase-like activity and was an effective antibacterial agent at low dose. As a result of the twin features of H_2_O_2_ production (oxidase-like behavior) and photothermal action, the SPB NCP seems to be exceptionally strong antibacterial agents.

#### 3.2.2. Chemical Induction Promotes Wound Healing

NO is a crucial regulating molecule. NO has been shown to modulate inflammation-related cytokines, have a positive effect on wound healing through a variety of different mechanisms, and reveal its major benefits including inflammation, vascular dilation, angiogenesis, cell proliferation, and matrix deposition during the inflammatory phase of wound healing [75,76,77]. Although nano carriers have been widely used in vivo and in vitro studies of NO carriers for bacterial and malignant tumors, the design and research of controlled wound healing based on the nano structure of NO release is still rare. Exogenous local delivery of NO is a promising method to enhance vasodilation and stimulate angiogenesis and collagen deposition. Su et al. synthesized PB nanocubes modified by Hemin for the local delivery of NO to the incision under near-infrared (NIR) light (Figure 7A) [78]. Using lithium aluminum hydride (LiAlH_4_) as a reducing agent, the cyanide (CN) group on the surface of PB nanocubes was reduced to -CH_2_-NH_2_ in the ice bath, and an amide bond was formed with the COOH group of Hemin. By thermally inducing NO release, colloidal PB nanocubes can be locally triggered to release NO multiple times at the wound site, thus promoting blood circulation, promoting angiogenesis, accelerating collagen deposition, and promoting wound healing. PB-NO colloids are easily dropped into the wound to be irradiated by near infrared light. However, it must be dropped into the wound several times. A single dosage is insufficient to offer a therapeutic impact, and the objective of improving blood microcirculation can only be achieved under controlled settings.

According to the in vivo skin wound healing experiment by Abhishek Sahu et al., the in vivo therapeutic effect of PBzyme was evaluated in a skin wound model [63]. The results show that in 50 μg × 4 groups, local injection of 50 μg PBzyme once every 4 days could accelerate wound healing, collagen deposition and tissue maturation. PB therapy can efficiently stimulate keratinocyte differentiation, enhance neovascularization, and decrease macrophage burden over the whole wound surface. Therefore, PBzyme not only accelerates the healing process in the simulated skin wound model, but also has the ability of scavenging ROS and anti-inflammation, and the tissue indeed shows regeneration properties. To calculate regeneration from the healing of infected wounds, photographs were taken on the wounds and the relative wound percentage was compared with blank at various time points over 14 days (Figure 7B). It has been observed from the beginning that PBzyme can accelerate wound closure. In vitro experiments showed that the optimal concentration of PBzyme for ROS scavenging was about 50 μg/mL. However, application of 50 μgPB in vivo did not yield satisfactory results. Nanozyme can be applied to traditional wound dressings for more practical or clinical applications. In conclusion, this study indicates that PBzyme is a promising nanomaterial for future wound healing and tissue regeneration.

Hao et al. synthesized NIR light-responded vancomycin-doped PBNPs (PB VANNPs) with highly efficient photothermal conversion for binding, dual-mode portable detection, and elimination of bacteria [42]. PB-VANNPs can bind to the surface of Gram-positive bacteria such as Staphylococcus aureus (*S. aureus*) to form PB-VANNPs/*S. aureus* (Figure 7C). Interestingly, during temperature-based detection, bacteria can be effectively inactivated as local temperatures rise, with excellent antibacterial efficiency. This portable and extremely sensitive nanoplatform innovates the way of “mixing before testing” while also detecting and removing bacteria, making the sterilization process more straightforward and efficient. The new approach was then used to improve wound healing in mice infected with bacteria.

There are issues to improve that include high cost, drug-related adverse effects, immunological rejection, and secondary harm to donor sites and wounds to enhance skin wound healing. Cao et al. produced a drug-free three-layered photothermal bactericide comprising copper sulfide (CuS), gold (Au), and ZnPBA (CuS@Au@ZnPBA) from the inside out [79]. CuS@Au@ZnPBA generates local heat under photothermal conditions, and the released Zn^2+^ can upregulate genes related to collagen deposition and promote bacterial ablation of wound infection in mice (Figure 7D). The long-term effects of CuS@Au@ZnpBA on tissues and tissue organs should be thoroughly explored before future utilization. Overall, well-designed nanocomposites have the promise to become an antibacterial alternative to current antibiotic treatments for bacterial wound infections.

### 3.3. Application of PB Nanoparticles in Antitumor

#### 3.3.1. Photothermal Tumor Therapy

PTT is a kind of local thermal therapy produced under the NIR light, which can be used for tumor ablation efficiently [80]. Tao et al. prepared novel Mn-doped PBNPs (Mn-PBNPs) by the microemulsion method, which demonstrated remarkable T1 and T2 weighted magnetic resonance imaging (MRI) improvement in a systematic experiment [35]. In addition, Mn doping enhanced the response of PBNPs to NIR and showed excellent Fenton reaction activity of PTT and chemo dynamic therapy (CDT) (Figure 8A). Overall, this Mn-PBNPs nanoplatform may be employed for multi-modal imaging, as well as CDT and moderate temperature PTT co-therapy, providing a dependable tool for tumor treatment. Similarly, Yu et al. developed a H_2_O_2_/NIR-responsive nanoplatform (MnPB @polypyrrole [MnPB@PPy]) to load Doxorubicin (DOX) by making MnPB@PPy core–shell-structured nanocarrier [81] for MRI-guided co-chemotherapy/hyperthermia. The results show that MnPB@PPy NPs can not only generate heat under NIR irradiation for cancer photothermal treatment, but are also excellent MRI contrast agents that have broad prospects for biomedical applications. Zhu et al. found that the longitudinal relaxation and NIR absorption of the doped PB nanocubes increased [82]. These features allow MnPB nanocubes with suitable surface coatings to ablate tumors successfully in vivo. The results show that Mn^2+^-doped PB nanostructures can be used as safe and effective nanoscale therapeutic drugs, and have great potential in future clinical applications.

PBNPs have high biosafety and blood compatibility, although it is currently stated that controlling the size below 50 nm is challenging. Smaller sizes are required for phagocytosis, circulation, and biological dispersion of cells throughout the body. To adjust the size of the PB probe, Shou et al. proposed zinc-doped PB nanoprobe (SPBZn (N%)) using an efficient and low-temperature aqueous solution reaction and produced the ultra-small PB probe (SPBZn (10%)) with a particle size less than 5 nm [36]. The results reveal that Zn^2+^-doping could produce ultrasmall PB nanoprobes and considerably increase their PTT and MRI performance, which is an effective medication for tumor detection and treatment and may be used in clinical tumor diagnostic and treatment applications (Figure 8B).

A pressing problem for tumor photothermotherapy is the discovery of suitable photothermal agents to fulfill complicated therapeutic demands. Li et al. created a platinum-doped PB nano-enzyme (Pt-PBzyme) with adjustable spectral absorbance and high photothermal conversion efficiency, as well as strong antioxidant catalytic activity, using a one-step reduction method [40]. The localized surface plasmon resonance (LSPR) frequency of Pt-PBzyme may be adjustable by altering the doping ratio, increasing photothermal conversion efficiency and allowing multiwavelength photoacoustic/infrared thermal imaging to guide PTT (Figure 8C). Due to its antioxidant catalytic activity, Pt-PBzyme also alleviates inflammation induced by hyperthermia. This discovery establishes a framework for developing safe and effective photothermal therapies to treat complicated tumor disorders.

Xu et al. synthesized uniformly dispersed Gd^3+^- and Tm^3+^-doped PB (Gd/TM-PB) by the solvothermal method [38]. By changing the Gd^3+^/Tm^3+^ ratio, Gd/TM-PB particles with the appropriate particle size and the best fluorescence and photothermal effects were obtained (Figure 8D). A multipurpose platform of Gd/TM-PB@ZIF-8/PDA for cancer diagnostics and therapy was constructed based on the optimization of Gd/TM-PB and the subsequent coating of polydopamine (PDA) functionalized MOF. Based on the optimization of Gd/TM-PB and the further coating of polydopamine (PDA) functionalized MOF, a multifunctional platform of Gd/TM-PB@ZIF-8/PDA for cancer diagnosis and treatment was established, which provided a new strategy for the integration of a nano-drug carrier in diagnosis and treatment.

Because of their unique structure, HPB nanospheres have a great drug load rate and NIR photothermal conversion efficiency. Lu et al. substituted Fe_2_O_3_ nanospheres for the conventional FeCl_3_ precursor, and then PBNPs were built into HPB nanospheres through an interface reaction [29]. Changing the growth period and template size under appropriate water and ethanol ratios allowed us to regulate the shell thickness and size of HPB nanospheres. The higher the concentration of HPB nanospheres was, the lower the survival rate of human cervical cancer cells (HeLa) was. The relative survival rate of HeLa was about 80% lower than that of cells without irradiation. Based on the findings, NIR irradiation can enhance the release of DOX from HPB nanospheres, which can combine PTT and chemotherapy to kill cancer cells. Therefore, the designed HPB nanosphere provides an ideal solution for cancer treatment due to its outstanding combination of NIR photothermal treatment and chemotherapy.

#### 3.3.2. Detection of Tumors and Cells

Dynamic monitoring of biological distribution and tumor targeting effect is particularly important to develop highly specific, highly sensitive and low-toxicity therapies, so to cure the primary tumor and prevent metastasis and recurrence.

To create SP94-PB-SF-Cy5.5, PBNPs with a porous metal organic frame loaded with sorafenib (SF) were coupled with the HCC-specific targeting peptide SP94 and the near-infrared dye cyanine (Cy)5.5 [83]. Multimodal imaging can be used to track the biodistribution and tumor-targeting effects of these nanoparticles, effectively removing tumor cells from the primary site through a combination of targeted, controlled SF release and favorable photothermal effects. For the first time, Akbal et al. produced folic acid doped PBNPs (FA-PB NPs) for theragnostic applications using the co-precipitation approach [39]. Typical characterization investigations reveal that the fa-Pb NPs modified sensor surface has a greater surface area, biocompatibility, and hydrophilicity, and is more likely to come into contact with cancer cells. This work demonstrates that the FA-PB NPs modified disposable sensor platform has a promising future in sensitive cancer cell detection.

Multiplex SERS detection of markers in tumor biosystems with no background provides an advantage over other optical approaches. Shen et al. proposed a quantitative classification technique for two kinds of breast cancer cells [84]. PBA coated gold nanoparticles (Au@PBA NPs) of the two types were devised and produced by replacing Fe^2+^ with Pb^2+^ or Cu^2+^. SERS nanoprobes could simultaneously and quantitatively detect the expression levels of MCF-7 and MDA-MB-231 biomarkers. Therefore, each sub-type can be described in a molecular spectroscopic manner by SERS emission based on the double C≡N bond, which is superior to typical flow cytometry methods. A sensitive and reliable miRNAs test is required for the early detection of non-small-cell lung carcinomas (NSCLC). Tang et al. showed a unique biocatalysis-mediated MOF-to-PB transformation (BMMPT) technique, as well as the catalytic hairpin assembly (CHA) signal enhancement methods [85]. It is based on PB@MOF-Fe^2+^ and tagged with a glucose oxidase (GOx) CHA amplification system. This creates a new approach for detecting both photothermal and electrochemical processes, with the purpose of improving the accuracy and convenience of miRNA detection in NSCLC.

Tracking the migration of various cells to different systems in vivo is essential to assess the effectiveness of immunotherapy. Zhang et al. demonstrated the good performance of Au@PB-Gd @ovalbumin NPs (APG@OVA NPs), which enables PBs to coordinate core with gold nanoparticles in cyanide (CN) binding for dendritic cell (DC) activation and labeling [86]. The APG@OVA NPs agents could not only activate and label DCs, but also monitor DC movement in real time and accurately profile DC distribution in the lymphatic system. APG@OVA NPs can be used as a high-performance tracer for DC immunotherapy due to its high activation effect, dual complementary imaging display and low biotoxicity. Zhao et al. developed a novel PB-doped reduced graphene oxide/MXene composite aerogel (3D PB/GMA) [87]. The electrochemical sensor based on 3D PB/GMA has strong electrocatalytic performance for H_2_O_2_ and a high pore volume and simulated peroxidation activity. When used to monitor H_2_O_2_ release from living cells in real time, it can distinguish cancer cell lines from normal cell lines, showing a great potential for application in pathological diagnostics. Mesenchymal stem cells (MSCs) can be combined with MRI to track transplanted stem cells, Wen et al. introduced PBNPs into the tracing of mesenchymal stem cells [88], and PBNPs-labeled MSCs demonstrated normal cell activity, migration, differentiation, and gene expression in controlled investigations, which demonstrated to be useful for identifying MSCs without compromising the biological properties of MSCs. This finding opens up new avenues for MSCs labeling.

#### 3.3.3. Load and Delivery of Antitumor Drugs

The drug delivery system (DDS) of biocompatible nanoparticles has brought broad prospects for the development of nanomedicine and has proved to be an ideal molecular carrier for diagnosis and therapy. It not only enhances the efficacy of targeted treatment medication delivery but also successfully cures malignant tumors [89,90,91]. DDS nanoparticles’ efficiency and efficacy are heavily influenced by their precise composition, surface characteristics, and shape. Among the published DDS nanoparticles, PBNPs are a novel and promising nano drug carrier due to unrivaled features such as high dispersion stability, ease of synthesis, low cost, and controlled shape in physiological environments [92]. The emergence of nanomedicine has given PBNPs new significance in the field of DDS. PBNPs are iron based nanoparticles and can be prepared in large quantities by simple methods at low temperatures [93]. Similar to iron oxide nanoparticles, PBNPs have proven to be a powerful contrast agent for MRI with the application of DDS, and their chemical stability, cytotoxicity and cell penetration have also been demonstrated [94]. In addition, the increase in PBNPs specific surface area can be achieved through structural control [95], thus increasing drug loading.

The recently developed NO-based gas treatment has been shown to lower multidrug resistance (MDR) and enhance absorption into tumor tissue. Fu et al. have innovatively doped sodium nitroprusside (SNP) into HPB nanoparticles as a precursor, which can produce NO and be used for drug delivery (NO-PB) [96]. The results demonstrated that the release rate of DOX NO-PB was faster in acidic conditions than in neutral pH conditions. NO-PB may boost drug-loaded drug release in acidic tumor tissues, decrease multidrug resistance, and improve penetration into tumor tissues. This opens a new way for nanocarriers to be used in combination therapy of NO and chemotherapeutic drugs as therapeutic agents.

So far, hypoxia in the tumor microenvironment promotes tumor growth and metastasis, leading to poor therapeutic efficacy. To solve this problem, Peng et al. constructed a nanocarrier named PBMn DOX@Red Blood Cell (RBC) (Figure 9A) [97]. PB/PBMn were used as oxygen precursors or catalysts activated by hydrogen peroxide to stimulate and activate drug release and alleviate hypoxia inside the tumors, overcoming the limitations of tumor microenvironment. RBC membrane was employed to boost DOX loading capacity, improve breast cancer chemotherapy/PTT, extend the in vivo circulation period, and decrease tumor development. These results suggest that PBMn DOX@RBC is a promising candidate for alleviating tumor hypoxia and enhancing chemotherapy/PTT.

In addition, protein, peptide and DNA drugs remain the frontier of PBNP-based DDS exploration. Currently, DNA drugs are considered as an emerging therapeutic molecule for the treatment of diseases such as infectious diseases and cancer [100]. In order to evaluate the killing effect and monitor the novel DNA nanodrugs on 22RV1 prostate cancer cells, Wang et al. employed PBNPs as a carrier to deliver DNA medications through DODN-MUA-PBNPs bound to cancer cells [98]. It was also discovered that the surface functionalization of PBNPs could not only transfer dODN to cancer cells but could also help with internalization and uniform distribution of dODN in the examined cells (Figure 9B). The killing power of cancer cells was boosted when the amount of dODN conjugated to PBNPs and the dose of DNA-PBNPs medication increased.

To reduce cancer mortality, the development of novel intellectual response systems optimizes co-administration and contemporaneous controlled release of several anticancer treatments. It will help reduce the probability of the evolution of treatment resistance. In this work, Chen et al. developed (PCM + Drugs) @ Hollow mesoporous PBNPs (HMPBs), an effective NIR light-sensitive medication co-delivery system based on HMPBs and PCM (1-tetradecanol) (Figure 9C) [99]. Under the NIR light, the photothermal reaction of HMPBs can cause the melting and escape of PCM, releasing the encapsulated pharmaceuticals, considerably reducing the release of hydrophilic DOX hydrochloride and hydrophobic camptothecin (CPT), and timely controlling the drug delivery switch, effectively inducing cell apoptosis. More significantly, it has a substantial photothermal-chemotherapy synergistic impact on HeLa cancer cells, which can boost drug absorption by enhancing cell metabolism and membrane fluidity. As a result, the produced (PCM + drug) @HMPB system shows promising potential as a NIR light response co-delivery platform for hydrophobic and hydrophilic anticancer medications, targeted adaptation drug resistance treatment, and collaborative photothermal-chemotherapy cancer treatment.

## 4. Conclusions and Future Prospective

In conclusion, the synthesis, characterization and role of PBNPs in regenerative medicine are reviewed. With the continuous development of nanomedicine, PBNPs have attracted increasing attention in preventing vascular restenosis, reducing excess reactive oxygen species and inflammation, while also promoting tissue repair, drug delivery and particle transport channels. However, at present, it is still limited to animal experiments, and has not yet been formally applied to the human body for clinical effect and systematic treatment. To overcome the constraints, in the future, researchers also need to explore through local injections of PB for clinical treatment in a nearly non-invasive way. In particular, PBzyme has an important reference value for clinical transformation (e.g., urethral stricture, reduction in scar formation, etc.) by describing its role in regenerative medicine and tissue engineering.

In general, PBzyme has unique advantages in preventing vascular stenosis, such as good dispersibility, low price and strong durability. It can be produced in large quantities with high enzyme activity. With these excellent properties, it is possible to treat long-term vascular restenosis. In reducing inflammation, PBNPs can be used as an artificial nano-enzyme to efficiently scour reactive oxygen species, develop new capabilities and improve current characteristics using complete substitution (PB analogue) and partial substitution (doping). In promoting tissue repair, PBzyme promotes blood circulation, angiogenesis and collagen deposition, thus accelerating the repair of skin tissue. In terms of drug loading and transport, PBNPs are an emerging nano-drug carrier with high chemical stability, low cytotoxicity and strong cell penetration. PBNPs increase the specific surface area through structural control, thereby increasing the drug load.

There are still some challenges. First of all, although the physicochemical properties, biosafety and hemocompatibility performance of PBNPs have been evaluated in detail in vitro and in vivo [15], the application of PBNPs in animal tissue penetration and cell uptake has not been sufficiently studied. Therefore, the clinical application of PBzyme has been controversial, and the ultimate goal has to apply PBzyme to clinical applications. Second, published data suggest PB have great promise as a treatment for inflammation. However, nano-enzyme drug research is still immature and currently rarely used in clinical trials. Preclinical studies require more information to determine the best timing and method of drug delivery. Finally, questions need further research with regard to the optimal size and concentration of PBzyme, animal models, injection dose, and long-term safety and efficacy. Nevertheless, with new medical technology and accumulation of clinical experiments, a better technical solution is in near future.

## Figures and Tables

**Figure 1 pharmaceuticals-15-00769-f001:**
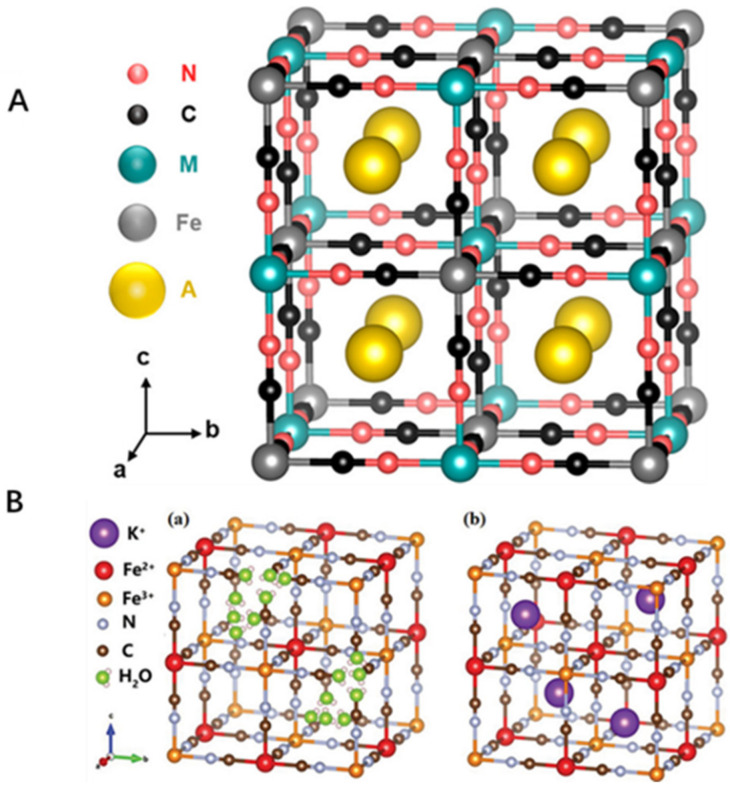
(**A**): Crystal structures of PBA unit cell. The general formula is A_2_MFe^II^(CN)_6_ (A = alkali metal ion; M = divalent transition metal ion) [18]. (**B**): The two main PB forms are: (**a**) insoluble PB crystal, (**b**) soluble PB crystal [14]. “Reprinted (adapted) with permission from [18] Copyright [18] American Chemical Society.” Copyright application has been approved.

**Figure 2 pharmaceuticals-15-00769-f002:**
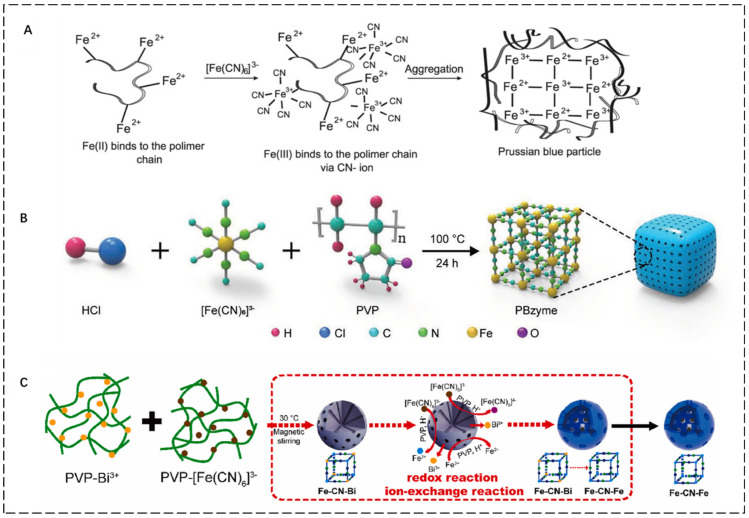
(**A**): Schematics of formation of the PVP−protected PBNPs [25]. (**B**): Schematics of the PBzyme synthesis process [26]. (**C**): Schematics of the synthesis of HPBzyme [27]. Copyright application has been approved.

**Figure 3 pharmaceuticals-15-00769-f003:**
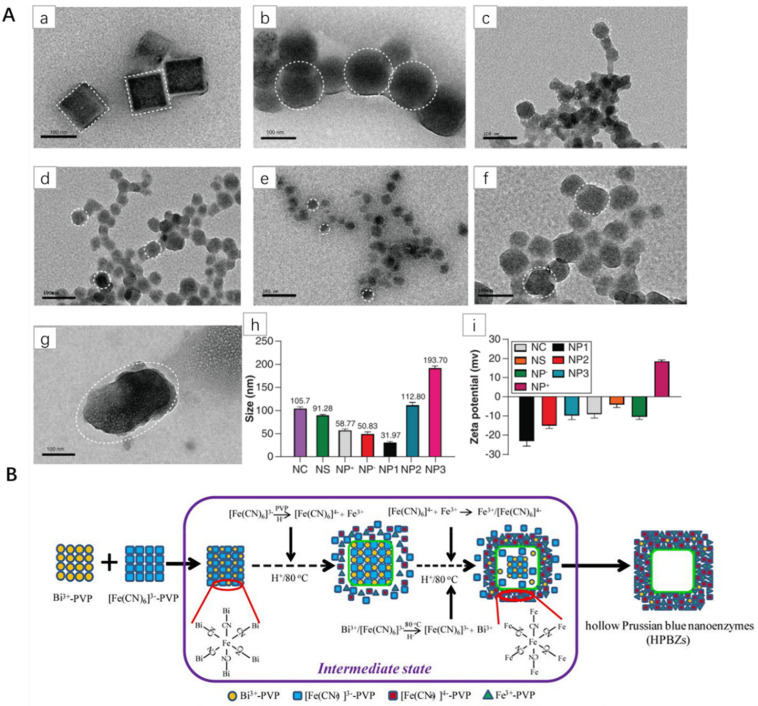
(**A**): Characterization of PBNPs. (**a**–**g**) Transmission electron microscopy indicated the shape and size of PBNPs; (**a**) NC, (**b**) NS, (**c**) NP+, (**d**) NP-, (**e**) NP1, (**f**) NP2, (**g**) NP3. (**h**) DLS size of PBNPs. (**i**) Surface charge potential of PBNPs. NC: Cuboidal PBNPs; NP: Nanoparticle; NS: Spherical PBNPs [15]. (**B**) Schematic diagram of HPBzyme formation [30]. Copyright application has been approved.

**Figure 4 pharmaceuticals-15-00769-f004:**
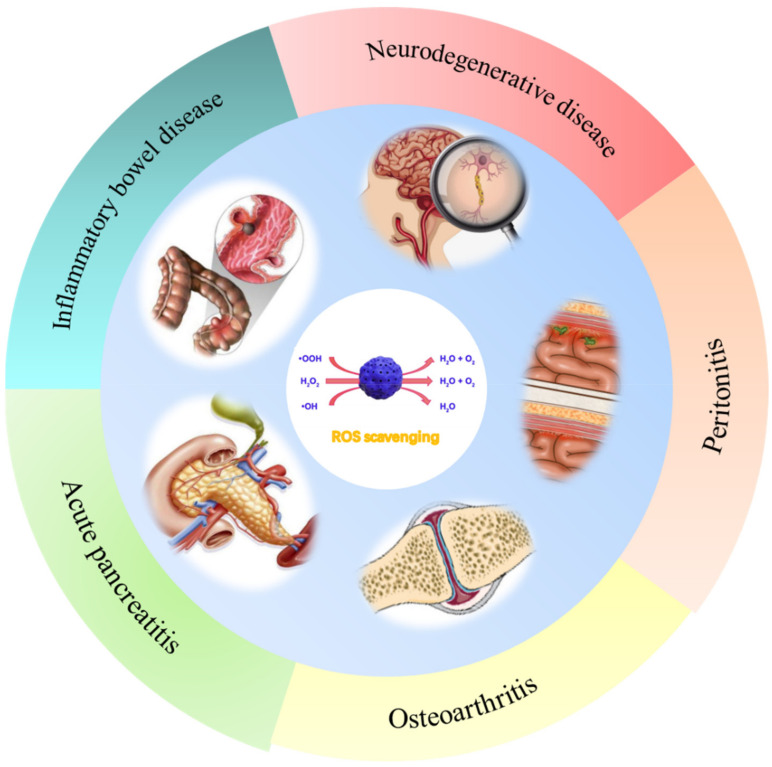
Summary of different applications of PBNPs in Acute pancreatitis, Osteoarthritis, Peritonitis, Neurodegenerative diseases, and Inflammatory bowel disease.

**Figure 5 pharmaceuticals-15-00769-f005:**
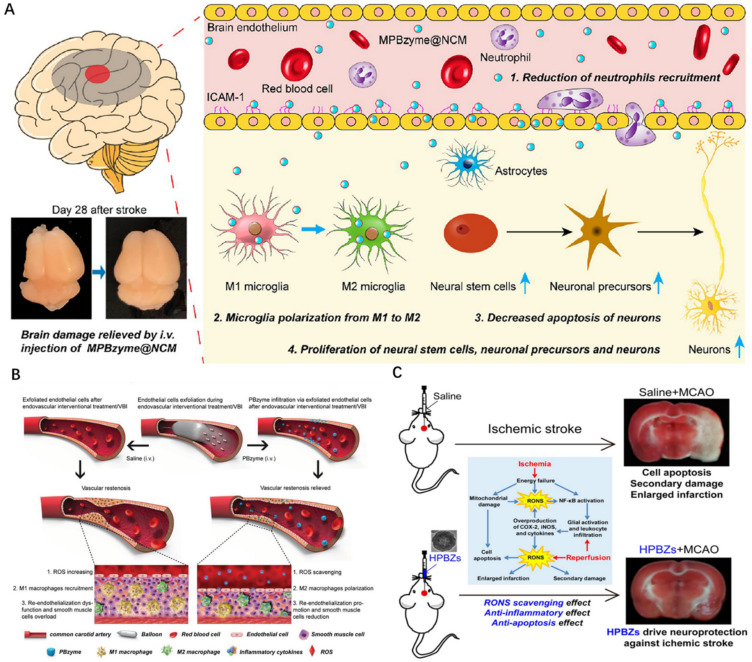
Schematic diagram of nano−enzymes preventing vascular restenosis. (**A**): MPBzyme@NCM detailed mechanisms for the treatment of ischemic stroke [47]. (**B**): Endothelial cell exfoliation was used to enhance the infiltration of PBzyme [26]. (**C**): Diagram of HPBzyme inhibiting apoptosis and inflammation [30]. Copyright application has been approved.

**Figure 6 pharmaceuticals-15-00769-f006:**
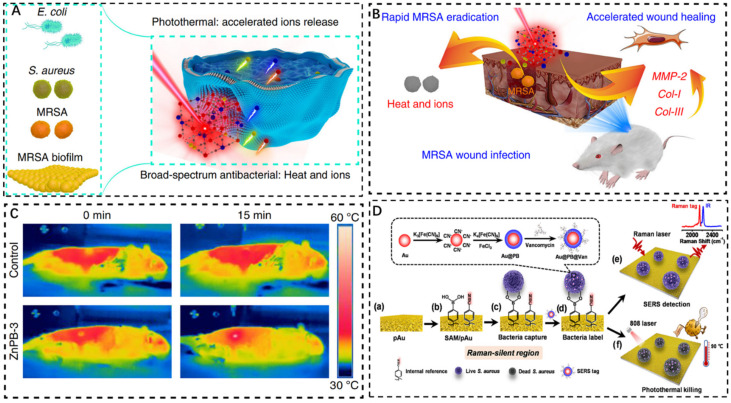
(**A**): Germicidal mechanism of ZnPB as exogenous antibacterial agent. (**B**): MRSA removal process in wound surface by ZnPB−3 photothermal treatment. (**C**): PTT comparison results of no NIR and znPB−3 irradiation for 15 min [72]. (**D**): Illustration of in situ photothermal sterilization of SERS platform [73]. Copyright application has been approved.

**Figure 7 pharmaceuticals-15-00769-f007:**
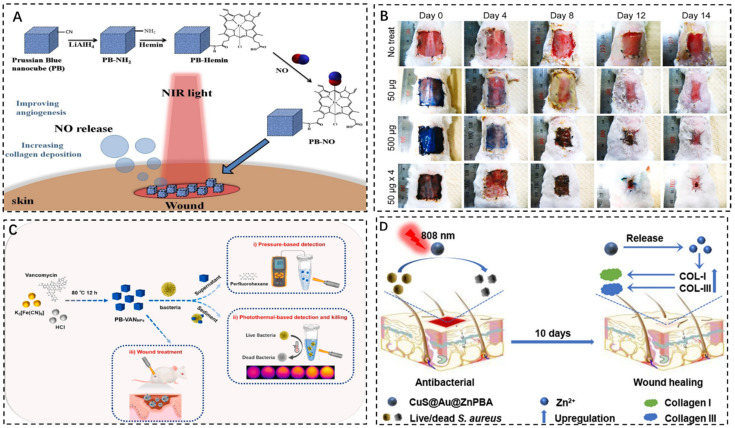
(**A**): PB nanocubes deliver NO to the wound site under NIR light [78]. (**B**): Observation of the effect of PBzyme in different groups on wound healing at different time [63]. (**C**): Schematic diagram of preparation, discovery, and removal of PB bacteria−VANNPs [42]. (**D**): CuS@Au@ZnPBA for photothermal sterilization and accelerated incision healing [79]. Copyright application has been approved.

**Figure 8 pharmaceuticals-15-00769-f008:**
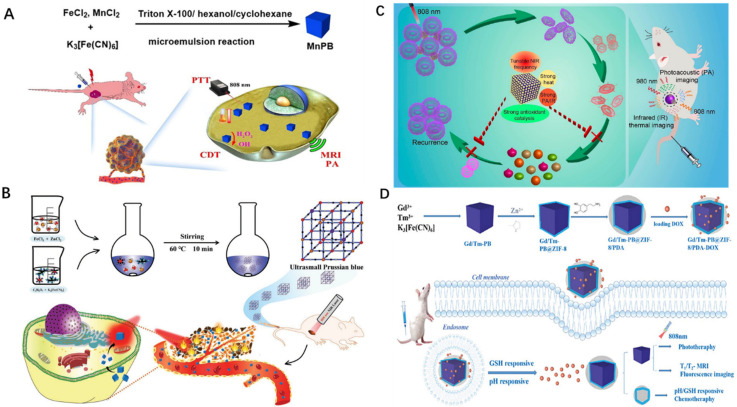
(**A**): MnPB NPs scheme for T1/T2 weighted MR imaging, PA imaging, and CDT/PTT co−therapy in cancer [35]. (**B**): Zn^2+^ doping to produce ultra−small PB nanoprobes with PTT and MRI characteristics [36]. (**C**): Pt-PBzyme as a photothermal agent to reduce inflammation and treat intricate tumor diseases [40]. “Reprinted (adapted) with permission from [40]. Copyright [40] American Chemical Society”. (**D**): The Gd/Tm−PB@ZIF−8/PDA platform with the best particle size, fluorescence, and photothermal effects [38]. Copyright application has been approved.

**Figure 9 pharmaceuticals-15-00769-f009:**
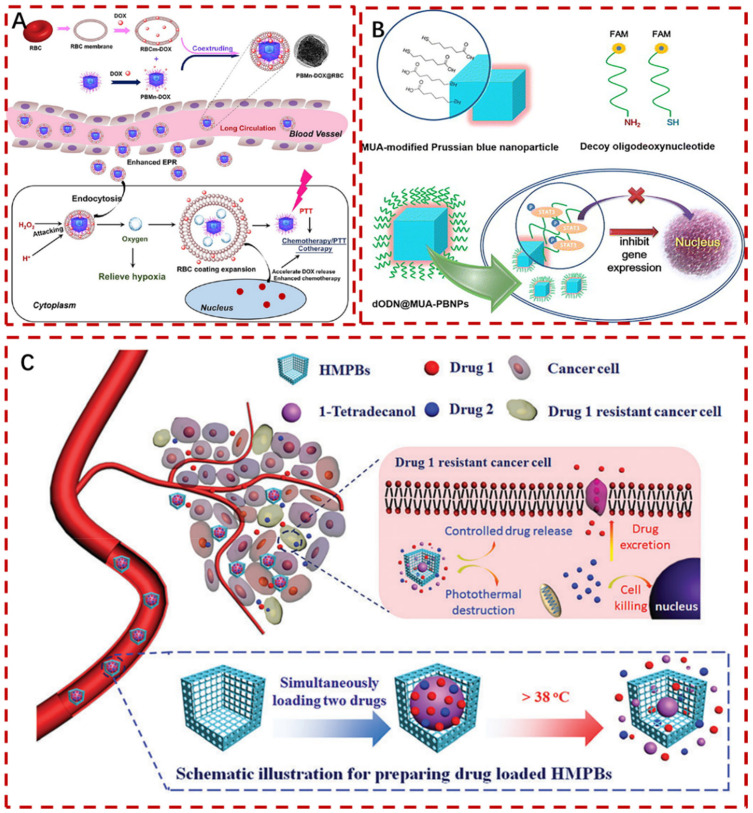
(**A**): Preparation of PBMn DOX@RBC nanocarriers and improvement of breast cancer chemotherapy/PTT to overcome tumor microenvironment limit [97]. (**B**): Schematic illustration of the dODN@MUA −PBNPs delivers drugs to cancer cells by coupling with DNA [98]. (**C**): (PCM+drug)@HMPB system as anticancer medications that are both hydrophobic and hydrophilic, in combination with photothermist−chemotherapy treatment of cancer [99]. Copyright application has been approved.

## Data Availability

Data is contained within the article.

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
