# Peer review of "Synthesis of Prussian Blue Nanoparticles and Their Antibacterial, Antiinflammation and Antitumor Applications"

_pharmaceuticals, 2022, doi:10.3390/ph15070769_

Round 1

Reviewer 1 Report

This manuscript entitled “Synthesis of Prussian blue nanoparticles and their application in antibacteria, anti-inflammation and antitumor” by D. Li. et al. described the different synthetic methods of Prussian blue nanoparticles and their application in biomedical application such as in tissue repair, anti-inflammatory and anti-cancer strategies. 

I think that the review could be published Pharmaceuticals if the following suggestions are taken into account.

1. The author must have the permission of the  respective journals to use the images of the figures presented here.

2. Change the quality of the Figure 3. It is impossible to clearly read the dimension of the size scale of the TEM images, the scale numbering on the plots, and the formulation of compounds that appear.

3. In Figure 4 - the authors should put figure description below corresponding figure. 

4. Ref 49 in section 3.1.3 - the authors argue that - "Since elevated levels of ROS affect cell proliferation and lead to apoptosis, excessive ROS accumulation at the wound site can lead to inflammation, necrosis and fibrosis scarring of skin cells, and ultimately delay the tissue repair process." Are there other studies that see the same effect?, support this with additional bibliography.

5. In section 3.2.1 - "Currently, the high temperatures of PTT have been effective in treating bacterial infections, but can also cause damage to healthy tissue" - Add bibliography that support it. 

6. Figure 6D - it is impossible to clearly read the text on it. 

7. Figure 7C - it is impossible to clearly read the text on it. 

8. Homogenize the word "anti-tumor" in all the text, that is, as antitumor or anti-tumor. 

9. The authors comments that "the average bond lengths of Fe(II)-N=C, C-N, and Fe(III)-N=C are 1.90, 1.13, and 2.03 Å, respectively (reference 3)". Check this out. I think the correct way would be like this: Fe(II)-C = 1.92 Å, C-N = 1.13 Å, and Fe(III)-N = 2.03 Å, respectively. Otherwise, add more references to support the value suggested. 

10. I suggest to change "anticancer medicines" for "anticancer treatments" or "anticancer drugs" as appropriate in the context. 

Reviewer 2 Report

The manuscript is written in review style.

The naming of the Prussion blue nanoparticles is changed several times, e.g. PBzyme, PBNP's etc.

Chapter 2.2: K3+ and K4+ - check abbrivation; there a plenty of other cases where abbreviations etc. have not be used correctly. Please check.

The largest part of the manuscript are in review style. Searching in a database several quite up-to-date and comprehensive reviews are found. Two examples:

The Application of Prussian Blue Nanoparticles in
Tumor Diagnosis and Treatment, Sensors 2020, 20, 6905; doi:10.3390/s20236905

Prussian blue nanoparticles: synthesis , surface modifications, and biomedical applicationsDrug discovery today, 2020, 25, 1431-1443; doi 10.1016/j.drudis2020.05.014

I believe the style of the manuscript, a quite short experimental part embedded in a review style manuscript is not appropriate. The new results they present are nicely fitting to a letter format.

In view of the missing balace between literature data and own scientific results I suggest to rewrite the manuscript either as a real review or as a scientific report with reduced literature citation.

Just one example: Fig. 9c is one-to-one taken from ref. 83. This is okay for a review but unusual in a scientific report paper.

At the time being I cannot support the publication of this manuscript.

Reviewer 3 Report

This review manuscript described the researches of Prussian blue nanoparticle, which possesses multi-enzyme activity, anti-inflammatory property, and scavenging reactive oxygen species. Particularly, it highlights many potent applications of PBNPs, including antibacterial, anti-inflammatory, antitumor, and drug carrier applications. Overall, this manuscript is informative and I suggest the following comments before the acceptance.

1.        Clearer description is sometimes required. For example, in 3.1.3, it is not clear on the correlation between MPBZ and ROS removal ability. It is better to provide an explanation of how PBNP synthesized using Mn.

2.        In Introduction, many references were not properly included, describing the benefits of PBNPs.

3.        Possible instability of PBNP should be further discussed. What is the possible strategy to circumvent this, in actual treatment?

Reviewer 4 Report

The authors present an interesting review about the potential uses of Prussian blue nanoparticles. The manuscript is well documented with an acceptable number of references, many of them very recent. It represents a good contribution to the field but the organization is a bit messy, mainly because the authors pretend to show all the information found without deepen in some parts as for instance the point 3.2.

PB medical applications are a little mixed up.

Figures are too complex in the sense that show excessive information. They difficult to see since they are not sharp enough.

Round 2

Reviewer 4 Report

The manuscript can be accepted in the present form.